# Self-association of MreC as a regulatory signal in bacterial cell wall elongation

Alexandre Martins[1,5], Carlos Contreras-Martel [1,5], Manon Janet-Maitre[1,2], Mayara M. Miyachiro[1], Leandro F. Estrozi [1], Daniel Maragno Trindade [3], Caíque C. Malospirito[3,4], Fernanda Rodrigues-Costa[3,4], Lionel Imbert[1], Viviana Job[1,2], Guy Schoehn [1], Ina Attrée[1,2] & Andréa Dessen [1,3✉]

The elongasome, or Rod system, is a protein complex that controls cell wall formation in rod-shaped bacteria. MreC is a membrane-associated elongasome component that co-localizes with the cytoskeletal element MreB and regulates the activity of cell wall biosynthesis enzymes, in a process that may be dependent on MreC self-association. Here, we use electron cryo-microscopy and X-ray crystallography to determine the structure of a self-associated form of MreC from *Pseudomonas aeruginosa* in atomic detail. MreC monomers interact in head-to-tail fashion. Longitudinal and lateral interfaces are essential for oligomerization in vitro, and a phylogenetic analysis of proteobacterial MreC sequences indicates the prevalence of the identified interfaces. Our results are consistent with a model where MreC's ability to alternate between self-association and interaction with the cell wall biosynthesis machinery plays a key role in the regulation of elongasome activity.

[1] Univ. Grenoble Alpes, CEA, CNRS, Institut de Biologie Structurale (IBS), Grenoble, France. [2] Univ. Grenoble Alpes, CNRS ERL5261, CEA-IRIG-BCI, INSERM UMR1036, Grenoble, France. [3] Brazilian Biosciences National Laboratory (LNBio), CNPEM, Campinas, São Paulo, Brazil. [4] Departamento de Genética, Evolução, Microbiologia e Imunologia, Instituto de Biologia, Universidade Estadual de Campinas (UNICAMP), Campinas, São Paulo, Brazil. [5] These authors contributed equally: Alexandre Martins, Carlos Contreras-Martel. ✉email: andrea.dessen@ibs.fr

Bacterial survival and morphogenesis are highly dependent on the peptidoglycan (PG), a key component of the cell wall. The PG is an essential heteropolymer that surrounds most bacterial cells, offering protection from osmotic lysis, and its biosynthetic machinery has been the target of successful antibiotics for decades. Proteins that are involved in PG biosynthesis associate in dynamic multi-membered complexes that regulate cell division (the "divisome") and cell wall elongation (the "elongasome", or Rod system), and their inhibition or deregulation can lead to defects in cell shape, impaired growth, and often cell wall lysis and death[1,2].

A key protein that regulates the positioning of the elongasome during the steps viewing incorporation of new PG along the lateral cell wall is the actin homolog MreB. MreB is largely conserved, and assembles in a nucleotide-dependent manner into short filaments associated with the inner side of the cytoplasmic membrane[3–8]. On the periplasmic side of the membrane, MreC has been shown to co-localize with the membrane-embedded and periplasmic components of the elongasome complex, including MreD, the monofunctional transglycosylase RodA, the transpeptidase Penicillin-Binding Protein 2 (PBP2), and RodZ. MreC is essential for shape maintenance in rod-shaped bacteria, and *mreC* knockout mutants are not viable[9–18].

MreC is a bitopic protein that harbors a large periplasmic domain, whose most noteworthy characteristic is a β−sandwich core reminiscent of 'butterfly wings'[18–20]. The core is buttressed by an N-terminal helix and a C-terminal Ala-Pro rich region (Fig. 1a, Supplementary Fig. 1), both presenting predictions of high flexibility. MreC organizes into patches or short filaments that have shown to co-localize and in some cases move in concert with MreB in different bacteria[3,7,9,15,16,21]. MreC has also been linked to regulation of elongasome activity through the activation of PBP2 and RodA[22] and interaction with MreD[23], in a process that could be dependent on MreC's ability to self-associate in an organized fashion in the cell. The molecular basis of this self-association capacity is unknown.

In order to explore this ability to self-associate, we employed electron cryo-microscopy (cryo-EM), sedimentation velocity analytical ultracentrifugation (SV-AUC), and X-ray crystallography to elucidate the structure of MreC oligomers from *Pseudomonas aeruginosa*. MreC$_{Pa}$ forms tubular assemblies composed of antiparallel protofilaments with its central, β-sandwich fold generating most of the interactions. Three highly conserved regions are essential for filament formation, with two of them playing key roles in tubular assembly in vitro. Finally, we show that these interaction regions also play a role in MreC's stability in vivo, further illustrating MreC's modularity and self-associating capacity that could serve as a regulatory signal in the bacterial cell wall elongation process.

## Results

**MreC from *P. aeruginosa* forms tube-like assemblies.** In order to explore MreC's ability to self-associate, negative-staining EM and sedimentation velocity analytical ultracentrifugation (SV-AUC) were employed to directly visualize and measure the hydrodynamic properties of the periplasmic forms of MreC from three rod-shaped bacteria, *P. aeruginosa* (MreC$_{Pa}$), *Escherichia coli* (MreC$_{Ec}$), and *Acinetobacter baumannii* (MreC$_{Ab}$). The sedimentation profiles of MreC$_{Ec}$ and MreC$_{Ab}$ displayed multiple peaks in a lower range, including 3.8 $s$ and 9.8 $s$ (MreC$_{Ec}$) and 8.8 $s$ (MreC$_{Ab}$), as well as peaks with much higher $s$ values (40.3 $s$ and 47.2 $s$ for MreC$_{Ec}$ and MreC$_{Ab}$, respectively). These forms generated heterogeneous patches as well as thin filaments that on occasion displayed a tendency to associate laterally, as seen on EM negative-staining images. MreC$_{Pa}$, which also presented

sedimentation peaks in comparable ranges (3.6 $s$, 8.0 $s$, and 44.5 $s$), was able to further associate into very large species (320 $s$, with a fitted $f/f_0 = 2.6$, which suggests a highly elongated shape) (Fig. 1b and Supplementary Fig. 2). On negative-staining EM grids, MreC$_{Pa}$ resembled short, curved filaments reminiscent of 'beads on a string', that, upon further concentration, displayed a tendency to self-associate laterally (Supplementary Figs. 3a–f).

Negative staining images of MreC$_{Pa}$ lacking the TM region and the first 21 residues of its N-terminal helix (MreC$_{Pa(36-330)}$) revealed organized bundles that associated into tube-like structures, some measuring several hundred nanometers (Fig. 1c). These structures, that presented three different diameters even within a single sample (220, 200, and 180 Å, Supplementary Fig. 4), were stable over wide ranges of pH and salt concentrations, and their formation was independent of the addition of cofactors or partner molecules. The 200 Å form, being the most widely represented, was thus further characterized by cryo-electron microscopy (cryo-EM; Fig. 1d).

**Cryo-EM and X-ray crystal structures of MreC reveal association into a polar filament.** 2D class averages showed an ordered internal structure (Supplementary Fig. 4), allowing the determination of the helical parameters. Subsequent three-dimensional (3D) helical reconstruction of the MreC$_{Pa}$ tubular assemblies showed that they are made of six sets of filaments consisting of two protofilaments organized in antiparallel fashion (green and cyan in Fig. 1e, f, 2a), with the core being formed by direct interactions between the 'butterfly' domains of laterally associated MreC subunits. The structure presents a 6-start helical arrangement with a 10.35 Å rise and 5.7145 (x2) MreC$_{Pa}$ subunits per turn. The final 3D reconstruction exhibits a global resolution of 3.5 Å (Supplementary Table 1); the central 'butterfly wings' are the most stable part of the tube-like structure while the initial section of the N-terminal helix is poorly defined in the cryo-EM map (Supplementary Fig. 4e). The slightly curved nature of the filaments is reminiscent of the shorter and thinner filaments observed by negative-staining EM in highly diluted samples (Supplementary Fig. 3a), which could represent sections that detached from the longer structures or strands that failed to associate laterally. In certain regions, 'unwinding' into sheets can be observed along the tubes.

In order to allow unambiguous tracing of the MreC$_{Pa}$ structure into the cryo-EM Coulomb potential map, we solved the high-resolution crystal structure of MreC$_{Pa(97-258)}$ that harbors only the central β-sandwich core region (Fig. 1a, Table 1). Fitting the refined crystallographic model of MreC$_{Pa(97-258)}$ into the cryo-EM map confirmed that within each protofilament, monomers are ordered in a polar, head-to-tail fashion (Fig. 2a), with the β-sandwich folds contributing with complementing charged regions towards formation of the interface between two protofilaments (Supplementary Fig. 5). Three main regions are involved in filament formation: Pro114/Phe115 (Region 1) and Glu188/Arg190 (Region 3), implicated in lateral interactions between the antiparallel protofilaments, and Arg175 (Region 2), involved in longitudinal (head-to-tail) packing within each protofilament (Fig. 2b).

**Protofilament-interacting regions are conserved and relevant in vitro and in vivo.** In order to evaluate the potential importance of these regions, we undertook the alignment of 3204 proteobacterial MreC variants (Fig. 2c). In what concerns Region 1, the hydrophobic character of amino acids in position A (Pro114 in MreC$_{Pa}$) is 75% to 94% conserved in α-, β- and γ-proteobacteria, whereas B (Phe115 in MreC$_{Pa}$) ranges from 29% in γ- to 71% in β-proteobacteria. The Arg residue (Region 2) is

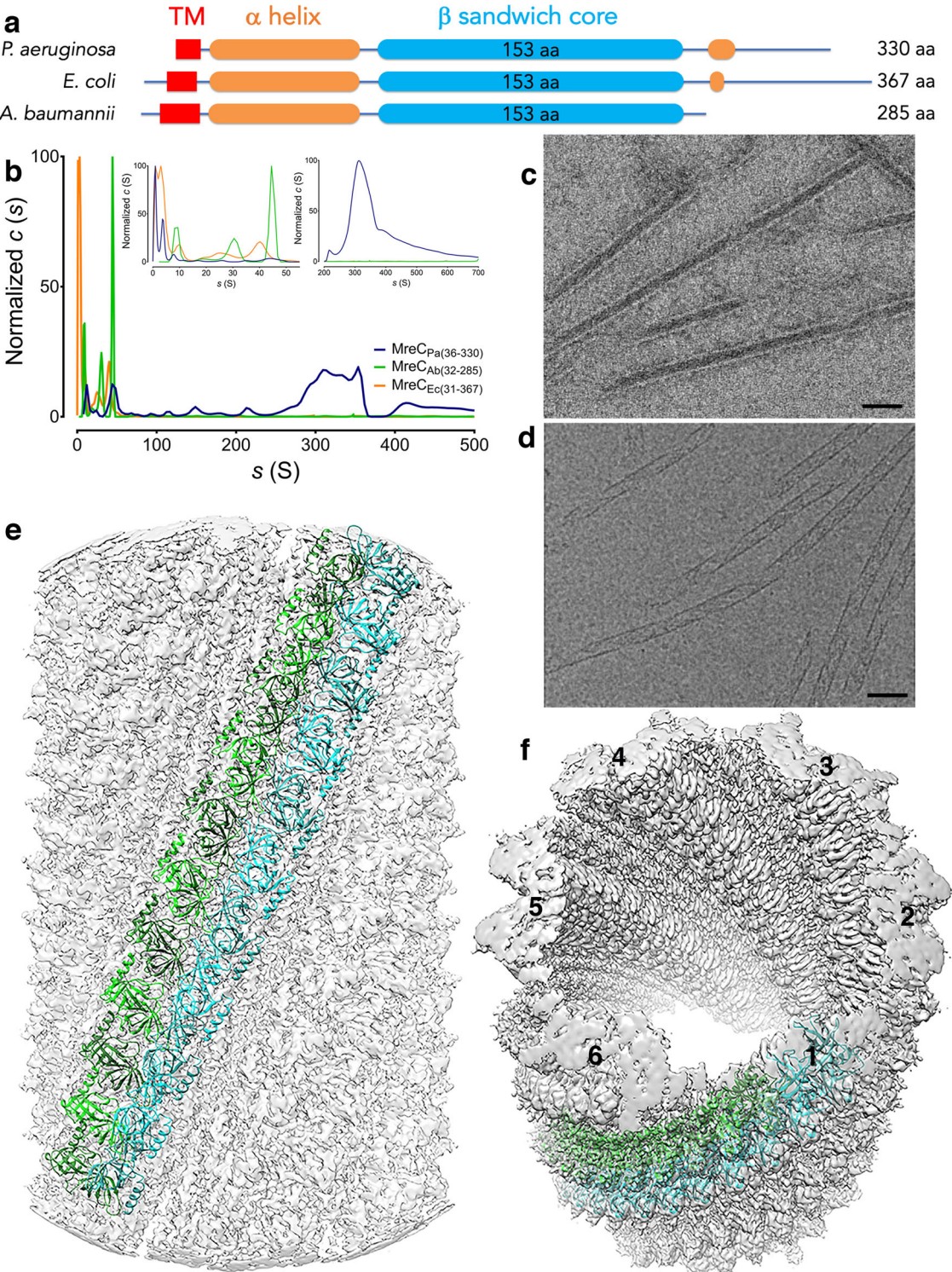

**Fig. 1 MreC$_{Pa}$ self-associates in vitro into bundles, tubes, and antiparallel filaments. a** Domain scheme of selected MreC variants. β-barrels were aligned to the 'butterfly' region of MreC from *P. aeruginosa* (the β-sandwich core). **b** SV-AUC sedimentation curves for MreC$_{Pa}$, MreC$_{Ec}$, and MreC$_{Ab}$. All three variants display very large *s* values in AUC, with MreC from *P. aeruginosa* showing the highest propensity to generate the largest oligomeric forms. Inset curves display different sedimentation ranges, with 0–50 (s) indicating the presence of oligomers from the three species, and the 200–700 (s) range indicating a pronounced peak for MreC$_{Pa}$ only. **c** Negative stain electron micrographs showing higher order structures formed by recombinant MreC$_{Pa}$ which were suitable for cryo-EM studies. The bars indicate 50 nm. During this study, over 300 images were made, and more than 20 grids prepared in total. **d** The same sample shown by cryo-EM reveals forms with diameters of 180 and 220 Å, and is a representative of images obtained from three different cryo-EM experiments, with 4 grids being prepared for the one highlighted here. **e, f** MreC$_{Pa}$ self-associates into tubes formed by six sets of antiparallel protofilaments. Individual protofilaments are shown in cyan and green. Inner and outer tube dimensions correspond to 150 Å and 280 Å, respectively. The tubular structures shown above represent the most stable in vitro form that allowed structural characterization. Source data are provided as a Source Data file.

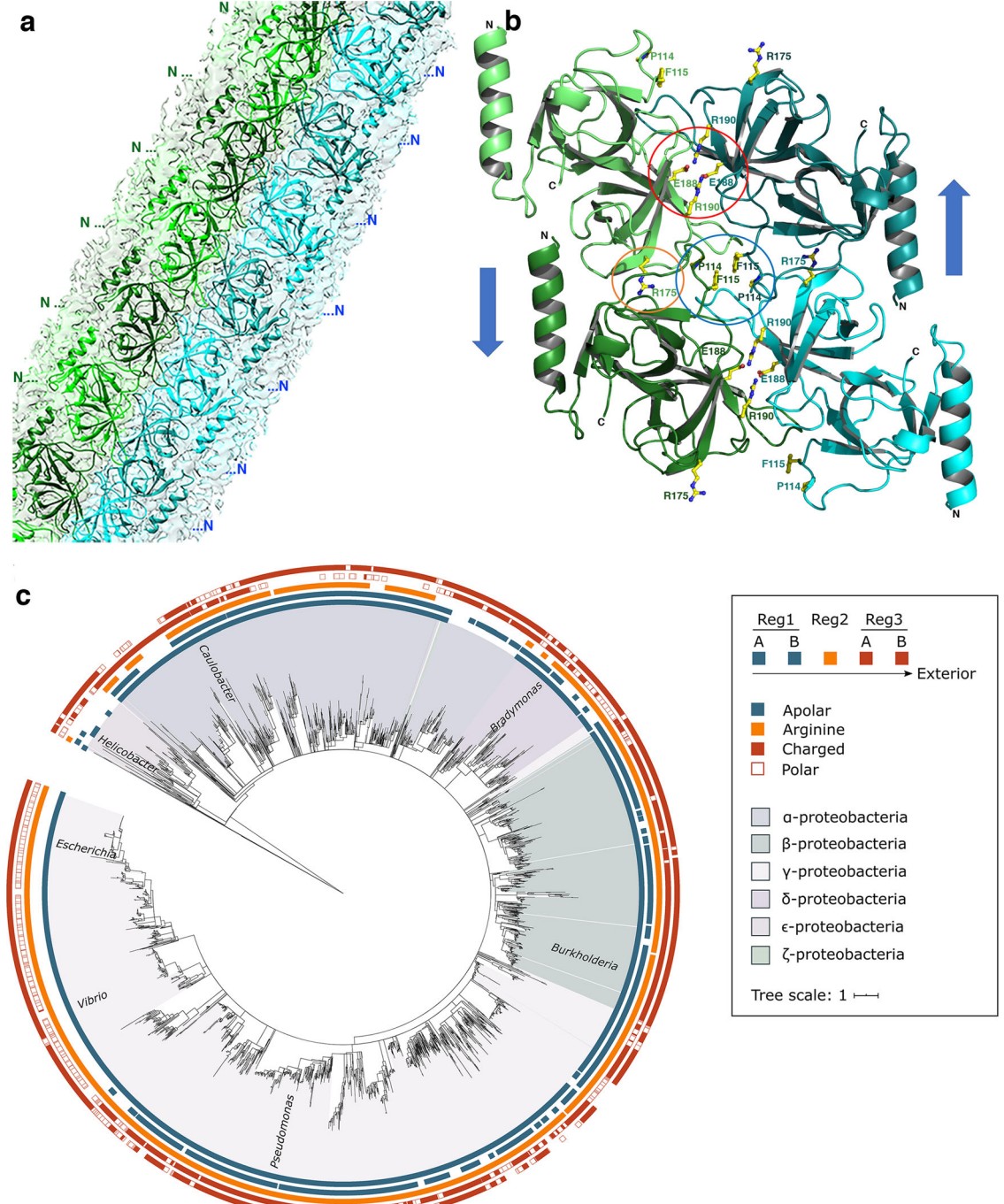

**Fig. 2 MreC$_{Pa}$ associates head-to-tail through three main conserved regions. a** Details of the cryo-EM map of an MreC filament; N-termini are aligned along each protofilament, with the flexible N-terminus pointing towards the outside. **b** The minimal repeating unit that displays all interfaces is shown as a tetramer for simplicity. Key residues studied by mutagenesis are highlighted as sticks. Regions 1, 2, and 3 are highlighted in blue, orange, and red circles, respectively. **c** Phylogenetic tree of MreC variants in Proteobacteria, highlighting the conservation of the three interaction regions studied in this work. Proteobacterial MreC sequences were aligned using PROMALS3D and a Maximum Likelihood tree was generated using MEGA X. Apolar amino acids (PIFAVLM) in Region 1 are depicted in blue, the Arg in Region 2 is shown in orange and charged (EDRK) and polar (QTSY) amino acids in Region 3 are shown as full or empty red squares, respectively. A and B indicate residue positions within Regions 1 and 3. Note that the highest level of conservation of the three key MreC regions lies within β- and γ-proteobacteria.

conserved in more than 95% of all variants in β- and γ-proteo-bacteria, and is the consensus in α- and δ-proteobacteria with 44% and 66% conservation, respectively. Finally, in Region 3, residues in positions A and B (Glu188 and Arg190 in MreC$_{Pa}$) are mainly polar and/or charged, separated by a hydrophobic residue. Overall, except in ε- and ζ-proteobacteria which seem to be

distinct, the three regions identified in this work are highly conserved throughout the proteobacterial phylum (Fig. 2c and Supplementary Fig. 1).

We sought to characterize the importance of the three different regions of MreC in vivo by introducing mutations (Region 1: P114G/F115A; Region 2: R175S; Region 3: E188A/R190G)

**Table 1 X-ray data collection and structure refinement statistics for MreC$_{Pa(97-258)}$.**

| DATA COLLECTION | |
| --- | --- |
| Data set | LNLS Campinas |
| X-ray source | LNLS-UVX |
| Detector | Pilatus 2 M |
| Wavelength (Å) | 1.45883 |
| Scan-range (°) | 360 |
| Oscillation (°) | 0.1 |
| Space group | P3$_1$21 |
| $a$ (Å) | 49.00 |
| $b$ (Å) | 49.00 |
| $c$ (Å) | 116.24 |
| Mosaicity (°) | 0.136 |
| Overall resolution (Å) | 42.44-1.47 |
| No. observed/unique reflections | 437496/27247 |
| High-resolution shell (Å) | 1.56-1.47 |
| Completeness (%) (last shell) | 96.1 (76.1) |
| $R_{sym}$ (last shell) | 4.1 (297.5) |
| $I/s(I)$ (last shell) | 25.90 (0.56) |
| CC1/2 (%) (last shell) | 100.0 (29.2) |
| Wilson plot B-factor (Å$^2$) | 38.28 |
| **ARCIMBOLDO Schredder** | |
| Final CC (%) | 26.27 |
| Residues Traced | 110 |
| **Arp/wARP** | |
| $R_{work}/R_{free}$ (%) | 25.39/29.40 |
| **REFINEMENT** | |
| Initial $R_{work}/R_{free}$ (%) | 25.39/29.40 |
| Final $R_{work}/R_{free}$ (%) | 21.68/24.45 |
| RMS deviation, bond lengths (Å) | 0.010 |
| RMS deviation, bond angles (°) | 1.332 |
| Mean B-factor (Å$^2$) | 39.46 |
| No. of protein/water atoms | 1178/66 |
| No. of Mg atoms | 1 |
| No. of Cl atoms | 5 |
| Residues in most favored/allowed region of Ramachandran plot (%). | 99.2 |

moderately anisotropic shape), suggesting that this region plays a minor role in the lateral association of MreC molecules and confirming what had been observed in the in vivo experiments. MreC$_{Pa-Region2}$ was unable to self-associate into any kind of protofilament or higher order oligomer as indicated both by the lack of fibrous structures in EM and the low $s$ values in SV-AUC (3.3 $s$, fitted $f/f_0 = 1.6$). Lastly, MreC$_{Pa-Region3}$ was still able to form high order oligomers with $s$ values of 41.2 $s$ and above in SV-AUC, but was unable to form tube-like structures as MreC$_{Pa(36-330)}$; instead, it generated thin fibers that seemed to be associations of two intertwined protofilaments, indicating that the mode of lateral association was distinct from that of MreC$_{Pa(36-330)}$ (Fig. 3). These data thus provided further evidence that the lateral and longitudinal interactions generated through Regions 2 and 3 have a clear effect on the ability of MreC$_{Pa}$ to self-associate, reflecting a potential mechanistic role within the *P. aeruginosa* elongasome.

## Discussion

In order to modulate lateral peptidoglycan biosynthesis, the elongasome must have a mechanism to sense whether all protein partners are correctly positioned. MreC was shown to interact with PBP2 and change its conformation into a state that in turn activates the glycosyltransferase activity of RodA[18,22,24]. These interactions are thus required for the synthesis of lateral peptidoglycan and wall elongation. However, the question remains as to how the cell turns peptidoglycan biosynthesis "off" once the appropriate cell length has been attained.

Superposition of our *P. aeruginosa* cryo-EM MreC$_{Pa}$ protofilament structure onto the only crystal structure of an MreC variant in complex with an elongasome partner, namely MreC from *H. pylori* (MreC$_{Hp}$) in complex with PBP2, sheds light onto this question (Fig. 4a). Formation of the PBP2-MreC$_{Hp}$ complex requires the "opening" of the N-terminal region of PBP2, allowing MreC$_{Hp}$ recognition[18]. The interaction region is formed by a 'hydrophobic zipper' involving a stretch of residues harbored by β7, β8, and β11 of MreC$_{Hp}$ (Fig. 4b). Superposition of MreC$_{Pa}$ onto the structure of the PBP2-MreC$_{Hp}$ complex reveals that the homologous interacting zipper in β7, β8 and β11 of MreC$_{Pa}$ is in close proximity to residues of Region 3, also located on β8 (in a range of 5–10 Å; Fig. 4c). This suggests that this region of MreC could be involved either in recognition of PBP2 (PG biosynthesis "on") or, in its absence, other MreC molecules (PG biosynthesis "off"), leading to its accumulation through the formation of higher order MreC oligomers, short filaments, or patches (Fig. 4d–f). The 'off' signal could be provided by MreD, which when bound to MreC and PBP2, has been suggested to prevent the conformational change of the latter[23]. This indicates that the different interacting surfaces of MreC could play a role in the regulation of the on/off states of peptidoglycan biosynthesis, potentially in addition to other interactions, such as allosteric sites in PBP2[25]. Indeed, ribosome profiling data from *E. coli* indicate that there is approximately twice as much MreC in the cell as PBP2 or MreD[26], suggesting that the oligomeric nature of MreC could be essential for regulation of the interaction between elongasome partners.

The MreC$_{Pa}$ forms studied here do not harbor the first 14 amino acids, encompassing cytoplasmic residues and the TM region, which, in the cell, participate in anchoring full-length MreC to the inner membrane. Nevertheless, in our cryo-EM structure, all N-terminal regions point to a common direction in each protofilament (Fig. 2a), indicating a pattern in which MreC molecules could be anchored side-by-side on the bilayer. Notably, a comparable arrangement has also been observed in crystal structures of other MreC variants, from *Streptococcus pneumoniae*, *Listeria monocytogenes*, and *H. pylori*, where the N-terminal

directly into the genome of *P. aeruginosa* by allelic exchange. In order to obtain a direct comparison of the effect of these mutations on MreC in the cell, we measured the amount of MreC in the three mutant strains both in exponential and stationary phases of cell growth (Supplementary Fig. 6). The strain carrying mutations in Region 1 displayed MreC levels that were comparable to wild-type *P. aeruginosa*, both in exponential and stationary phases. However, strains carrying mutations in Regions 2 and 3, despite displaying normal cell growth patterns in laboratory conditions, showed diminished levels of MreC in both phases, and most notably in stationary phase. These observations indicated that residues within MreC's Region 1 could be modified without detriment to the cell, whilst suggesting that those involved in Regions 2 and 3 have an effect on MreC's stability within the elongasome. This pointed to the possibility that the lateral and longitudinal interactions involving Regions 2 and 3 could play an important role in MreC's functionality in the cell.

We thus expressed and purified variants of MreC$_{Pa}$ where each interacting region was mutated accordingly (P114G/F115A, MreC$_{Pa-Region1}$; R175S, MreC$_{Pa-Region2}$; E188A/R190G, MreC$_{Pa-Region3}$). Mutant proteins were characterized by negative-staining EM and AUC (Fig. 3 and Supplementary Fig. 7). MreC$_{Pa-Region1}$ was able to form tube-like structures and large, elongated oligomers that resembled those identified for MreC$_{Pa(36-330)}$ ($s = 40.2$ with a fitted $f/f_0 = 1.4$, compatible with a

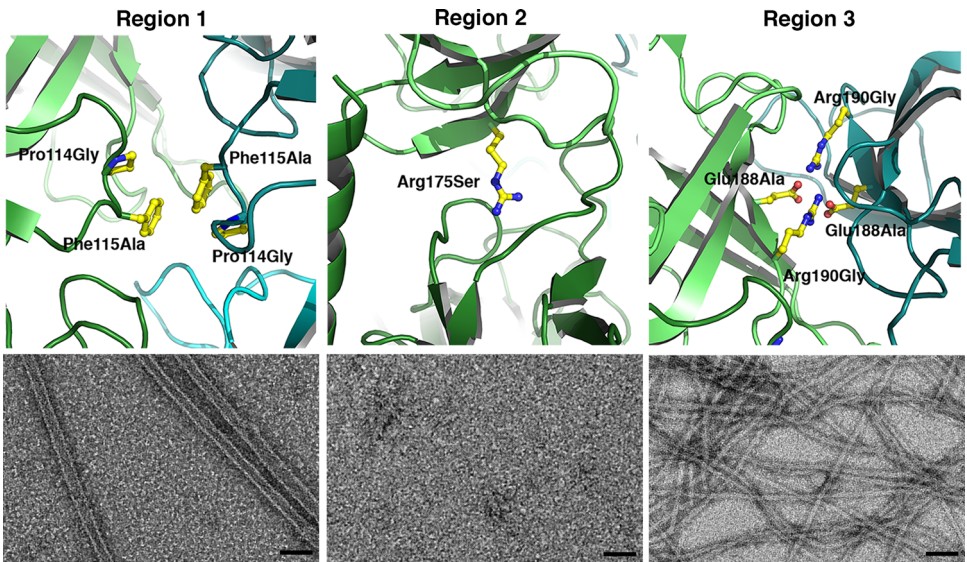

**Fig. 3 Lateral and longitudinal interactions play key roles in MreC association.** (left) MreC$_{Pa-Region1}$ is able to form tube-like structures that resemble MreC$_{Pa(36-330)}$ (4 grids, 42 images). (center) MreC$_{Pa-Region2}$ does not generate any large oligomeric forms that can be visualized by negative-staining EM (4 grids, 39 images), and SV-AUC (Supplementary Fig. 7) indicates a sedimentation profile that is indicative of a much smaller species. (right) MreC$_{Pa-Region3}$ forms thin fibers that are distinct from those observed for MreC$_{Pa(36-330)}$ (9 grids, 80 images). Scale bars in EM images correspond to 50 nm.

regions all point in a similar direction[18–20]. Nevertheless, structures presented in this work were performed with purified proteins that assembled into oligomeric species in vitro, and thus it is conceivable that differences could arise in the context of the native environment of the cell. The flexible nature of the N-terminal helix (which could only be partly traced in the cryo-EM map) could facilitate local rearrangements of MreC as it recognizes PBP2 (and/or other periplasmic elongasome partners) during the on/off stages of PG elongation. It is tempting to propose that these modifications could also be sensed by cytoplasmic members of the elongasome (such as MreB) through interactions transmitted through MreC's cytoplasmic and TM regions. This model reinforces recent suggestions that the multi-enzyme complexes involved in PG synthesis are dynamic, being able to assemble and disassemble depending not only on the stage of the cell cycle but also on the condition of the periplasm, which can be affected by the cellular environment (including local pH and osmolality[27]). Given the key nature of MreC and its sequence and structural similarities in rod-shaped bacteria (Fig. 2c and Supplementary Figs. 1 and 8), a regulation mechanism involving control by the self-association of MreC (Fig. 4f) could play a common role in elongation of the bacterial cell wall. The structure of the MreC oligomer described here offers a molecular overview into how this regulation can occur and provides further insight into the functioning of the bacterial elongasome.

## Methods

**Cloning of MreC variants.** *mreC* genes were amplified by PCR either from genomic DNA (for *P. aeruginosa* PAO1 and *E. coli* BL21(DE3)) or from a synthetic construct (*A. baumannii* AB030) purchased from Invitrogen/Thermo Fisher Scientific. The *A. baumannii* construct (strain NCBI accession number NZ_CP009257.1) was codon optimized for *E. coli* expression. All *mreC* variants were cloned using *Bam*HI/*Xho*I in a pGEX-4T1 vector in frame with the sequence coding for an N-terminal GST tag. All truncations and residue point mutations were created by site-directed mutagenesis with primers designed using the NEBaseChanger tool (http://nebasechanger.neb.com/). For mutagenesis of MreC$_{Pa(36-330)}$, primers were phosphorylated and used for the amplification reaction. The reaction product was incubated with *Dpn*I (Thermo/Fermentas) to eliminate the template, subsequently purified from an agarose gel and then ligated and transformed into MACH1-T1R competent cells. Constructs were confirmed by sequencing, and all primers and clones are described in Supplementary Table 2.

**Expression and purification of MreC variants.** Vectors expressing MreC variants from the three species were transformed into *E. coli* BL21(DE3) Gold cells (Novagen) and grown at 37 °C in LB liquid medium supplemented with ampicillin at 100 μg/ml. When the absorbance at 600 nm reached 0.6 A.U., protein expression was induced by the addition of 1 mM IPTG. Growth was continued overnight at 20 °C. The cell pellet was resuspended in Buffer A (25 mM CHES pH 9.0, 500 mM NaCl, 10 mM MgCl$_2$, 5% glycerol) and the cells were disrupted by using a cell disruptor (Constant Systems). After centrifugation, the soluble fractions containing the GST-tagged MreC forms were purified over a GST column and the protein was eluted in Buffer B (Buffer A + 20 mM glutathione). Eluted protein was injected in Superdex 200 16/600 (GE), and peak fractions were collected. In order to remove the GST tag, 5 units of thrombin were added per 1 mg of protein. Samples were incubated for 4 h at 4 °C. A reverse GST affinity was performed to collect the GST-free protein.

**Negative staining electron microscopy.** Prior to grid analysis, samples were concentrated on a Vivaspin concentrator in a buffer containing 25 mM CHES pH 9.0, 150 mM NaCl, and 10 mM MgCl$_2$. Negative-staining grids were prepared using the mica-carbon flotation technique[28]. Samples were adsorbed on the clean side of a carbon film previously evaporated on mica and then stained using 2% (w/v) Sodium Silico Tungstate pH 7.4 for 30 s. The sample/carbon ensemble was then transferred to a grid and air-dried. Images were acquired under low dose conditions (<30 e⁻/Å$^2$) on a Tecnai 12 FEI electron microscope operated at 120 kV using a Gatan ORIUS SC1000 camera (Gatan, Inc., Pleasanton, CA).

**Cryo-EM.** Three different cryo-EM experiments were performed. For the initial ones, the concentration of the sample did not allow the collection of enough data to perform image analysis. However, tube morphology was the same as what had been observed in negative staining. For the last experiment, four different grids, where sample concentration was appropriate, were frozen. We chose the best grid, taking into consideration the number of squares and holes with the appropriate ice thickness.

Quantifoil grids (300 mesh, R 1.2/1.3) were negatively glow-discharged at 30 mA for 45 s. Multiple blotting was used to increase the concentration of filaments on the grid: twice in a row, 3.5 μl of the sample were applied onto the grid, and excess solution was blotted away by hand using Whatmann 4 filter paper. For the last step 3.5 μl of the sample were again applied on the grid but this time blotted away with a Vitrobot Mark IV (FEI) (blot time: 6 s, blot force: 0, 100% humidity, 20 °C), before plunge-freezing in liquid ethane. The grid was transferred onto a 200 kV Thermo Fisher Glacios microscope equipped with a Falcon II direct electron detector for data collection.

Automated data collection was performed with EPU, acquiring one image per hole, in counting mode. Micrographs were recorded at a nominal ×120,000 magnification giving a pixel size of 1.206 Å (calibrated using a β-galactosidase sample) with a defocus ranging from −0.8 to −3.5 μm. In total, 1200 movies with 20 frames per movie were collected with a total exposure of 43 e⁻/Å$^2$.

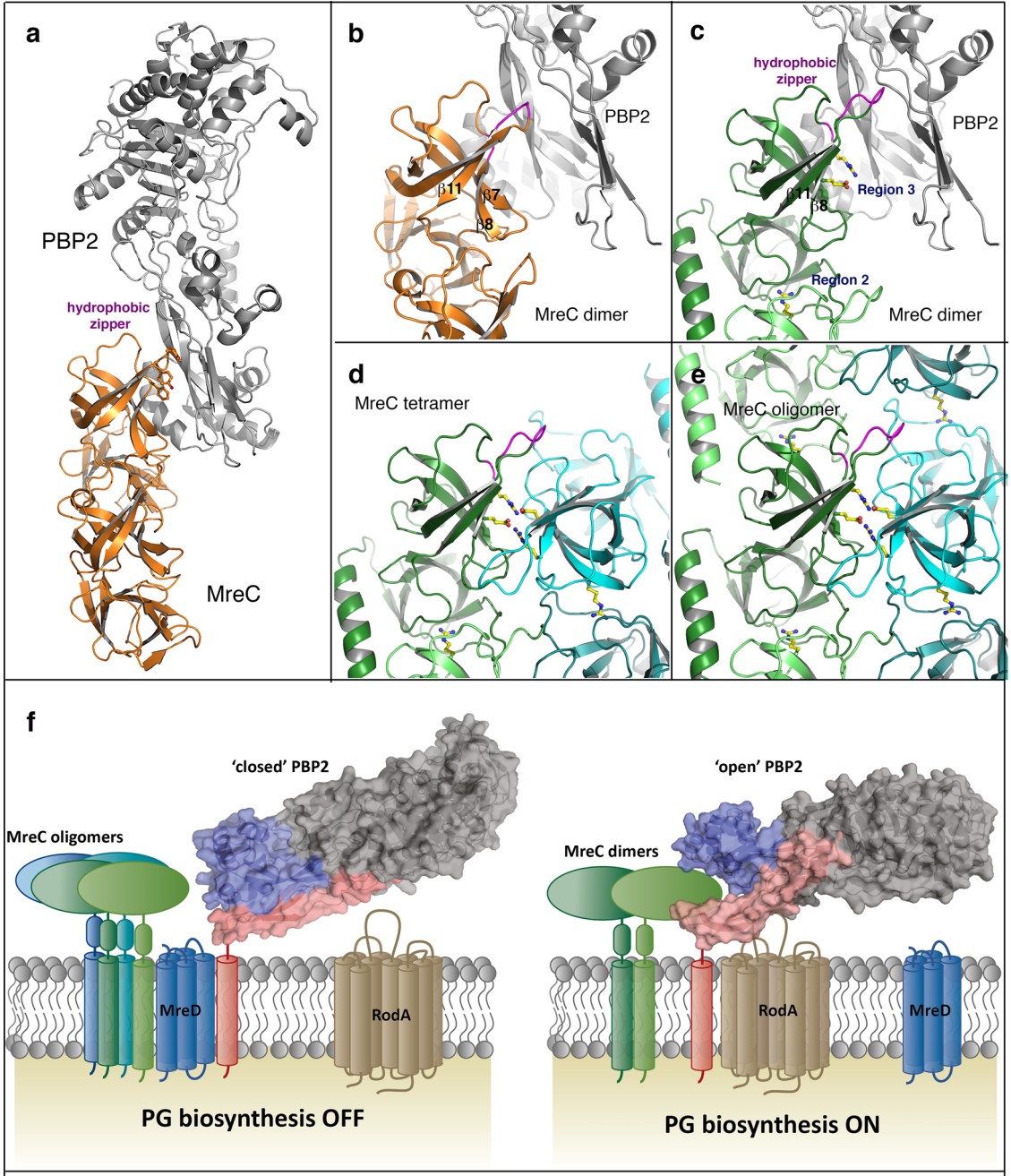

**Fig. 4 MreC modulates PG formation during cell wall elongation by self-association. a** Crystal structure of the MreC:PBP2 complex from *H. pylori* (PDB 5LP5). **b** closeup of the MreC:PBP2 interaction region, with the hydrophobic zipper region of MreC highlighted in magenta. MreC$_{Hp}$ is in orange, PBP2 in gray. **c** overlay of the MreC$_{Pa}$ dimer (green) onto MreC$_{Hp}$ within the MreC:PBP2 structure reveals that the potential hydrophobic zipper of MreC$_{Pa}$ is in close proximity to Region 3 (Glu188, Arg190). In this situation, MreC's self-association capacity is blocked by the presence of PBP2. **d, e** in the absence of PBP2, Region 3 is free and MreC dimers associate into stable tetrameric forms, giving rise to higher-order structures that do not interact with PBP2. Arg175 (Region 2) is essential for interactions both in the presence and absence of PBP2. **f** Schematic model of MreC's modulation of cell wall elongation. PBP2 is shown in gray, pink (anchor) and blue (head). MreC oligomers and PBP2, both contacted by MreD[23], do not interact and PG biosynthesis is off. A signal, which could involve accumulation of MreC beyond a given threshold, displaces MreD and allows MreC to interact with an open form of PBP2, which in turn recognizes RodA and activates PG biosynthesis[22].

**Image processing and cryo-EM structure refinement**. Movie drift correction was performed with Motioncor2[29] using frames from 2 to 19. CTF determination was performed with Gctf[30] and the obtained defocus values were within the 1.0–3.5 μm range. 739 movies out of 1200 were kept at this stage. 2,530 filaments were picked manually giving a total of 111,624 segments followed by 2D classification with Relion[31]. 2D classification showed that 88% of the segments were representative of the form that was selected for structure solution, while only 3.3% represented the 180 Å and 220 Å diameter forms. An initial 3D model was

created by the SPRING program[32] based on the best 2D class averages and the helical symmetry parameters were determined with the help of Helixplorer (Supplementary Fig. 4, http://rico.ibs.fr/helixplorer/). All subsequent image processing steps (3D refinement, polishing, post-processing, Fourier-shell correlation, local resolution estimation) were performed with Relion. In the final reconstruction (3.5 Å resolution, 0.143 FSC threshold), 82% of the original 111,624 segments were included (91,840). The remainder of the particles displayed problems such as tube deformation or poor contrast, and also included

particles with other diameters (3.2% of 111,624). At this point, the crystallographic model of MreC$_{Pa(97-258)}$ was introduced and refined with CCP-EM[33,34] and COOT[35], employing the same strategies as described below for crystal structure solution.

**MreC$_{Pa(97-258)}$ crystallization, data collection, and structure solution**. A clone expressing just the 'butterfly' region of MreC (residues 97–258), identified through sequence alignments using the sequences of MreC variants whose structures were available[18-20] was employed for X-ray crystallography efforts. MreC$_{Pa(97-258)}$ was expressed and purified as described above for other variants, with the exception of the buffer employed (HEPES pH 8.0) and thrombin cleavage details (performed directly on the column). The cleaved protein was further purified by size exclusion and the single peak was concentrated using a Vivaspin concentrator. Crystals were grown by the vapor diffusion method at 18–20 °C using a hanging-drop setup. MreC$_{Pa(97-258)}$ was crystallized by mixing equal volumes of protein sample (10–11 mg/ml in 20 mM HEPES, 200 mM NaCl) and of reservoir solution (100 mM imidazole pH 6.5, 1.5 M NaCl, 15% w/v PEG 3350, 100 mM MgCl$_2$). Crystals were cryoprotected with Parabar 10312 (Hampton Research), mounted on cryo-loops and flash-cooled under liquid nitrogen.

Two data sets were collected; an initial, in-house set that diffracted to 1.74 Å, and a higher resolution (1.47 Å) data set collected on beamline MX2 at the LNLS synchrotron in Campinas, Brazil. X-ray diffraction images were indexed and scaled with XDS[36]. ADXV (http://www.scripps.edu/tainer/arvai/adxv.html) and XDSGUI (https://strucbio.biologie.uni-konstanz.de/xdswiki/index.php/XDSGUI) were used to perform data quality and resolution cutoff verifications. The maximum possible resolution was determined using the STARANISO server (http://staraniso.globalphasing.org/cgi-bin/staraniso.cgi). The reduced X-ray diffraction data were imported into the CCP4 program suite[37]. ARCIMBOLDO[38] was deployed in a local HTCondor (v8.6.6) cluster in order to solve the structure by direct ab initio phasing methods. Generation of a template list of distant homologs was achieved using the HHpred server[39]. Each one of the templates was then tested with ARCIMBOLDO_SHREDDER module in Spherical Mode[40]. The first automatic model re-building with ARP/wARP[41] docked 60 residues of the sequence using the 1.74 Å data set. The structure was completed by cycles of manual model building with COOT by phase extension using the data collected to 1.47 Å. Crystallographic macromolecular refinement was performed with REFMAC[42]. Water molecules were added to the residual electron density map as implemented in ARP/wARP and COOT, and cycles of manual model building and refinement were performed until $R_{work}$ and $R_{free}$ converged. The TLS definition[43] was determined and validated using the TLSMD and PARVATI servers[44,45]. The stereochemical quality of the refined models was verified with MOLPROBITY[46], as implemented in COOT, and PROCHECK[47]. X-ray diffraction data, structure solution and refinement statistics are found in Table 1. Figures displaying protein structures were generated with PyMol 1.7. (http://www.pymol.org).

**Sedimentation velocity analytical ultracentrifugation (SV-AUC)**. MreC$_{Pa(15-330)}$, MreC$_{Pa(36-330)}$, MreC$_{Pa-Region1}$, MreC$_{Pa-Region2}$, MreC$_{Pa-Region3}$, MreC$_{Ec}$, and MreC$_{Ab}$ at 0.8 mg/ml were analyzed by SV-AUC in order to obtain information regarding the shape and oligomerization state of each protein. 12 mm 2-channel Ti centerpiece cells (Nanolytics, Potsdam, DE) (400 μl) were employed. Cells were filled with a control buffer (25 mM CHES pH 9.0, 150 mM NaCl, 10 mM MgCl$_2$) in the reference sector and with the sample solution in the sample sector. Sedimentation velocity experiments were performed using a XLI ultracentrifuge (Beckman Coulter, Palo Alto, USA) and an 8-place Anti-50 Ti Analytical Rotor (Beckman Coulter) at 62,000 $g$ and 20 °C with absorbance monitoring at 280 nm. Data were processed with Sedfit[48]. The analysis was performed through the continuous size distribution c(s) method to determine the values of the sedimentation coefficients, $s$. In this method, a frictional ratio, $f/f_0$, representing the mean shape and hydration for all sedimenting macromolecules, is fitted. A value of $f/f_0 = 1.25$ corresponds to a globular compact shape, while larger values reflect anisotropic or elongated shapes. The $s$ values in this study are reported in Svedberg (S) units, which correspond to $10^{-13}$ s. The partial specific volumes of each sample were calculated from protein sequences in Sednterp (http://sednterp.unh.edu)[49].

The molecular parameters that determine the $s$-value of the oligomeric states are given by Svedberg equation:

$$S = M(1 - \rho\bar{v})/N_A 6\pi\eta R_h$$

where $M$ and $\bar{v}$ are the molar mass and the partial specific volume of each protein. $N_A$ is Avogadro's number. The hydrodynamic radius ($R_h$) is related to $R_{min}$, the radius of the anhydrous volume ($\bar{v}M/N_A$), and to the frictional ratio $f/f_0 = R_h/R_{min}$.

**Construction of *mreC* chromosomal mutants**. DNA fragments of the *mreC* gene were amplified by PCR from the genomic DNA of *P. aeruginosa* PAO1, and the desired nucleotide substitutions leading to Pro114Gly/Phe115Ala (Region 1), Arg175Ser (Region 2) and Glu188Ala/Arg190Gly (Region 3) changes were introduced by site-directed mutagenesis. For each mutation a new restriction enzyme site was introduced in order to verify the presence of the mutation in the genome (for Region 1: *Nar*I, Region 2: *Dde*I and Region 3: *Sty*I). Briefly, fragments were integrated into a pEXG2[50] vector by Sequence- and Ligation-independent Cloning (SLiC)[51] and the desired changes were introduced using a QuikChange II kit (Agilent) and further amplified in TOP10 *E. coli* (Invitrogen). Triparental mating using pRK600[52] was used to transfer the mutated fragments to the *P. aeruginosa* chromosome. Merodiploids were selected on LB plates containing irgasan (25 μg/ml) and gentamicin (75 μg/ml). Single colonies were then re-streaked on NaCl-free LB agar plates supplemented with sucrose 10% (w/v) to select for plasmid loss and double recombinants[53]. Resulting clones were verified for gentamicin sensitivity indicating plasmid loss. Screening for desired mutants was performed by PCR of the desired region, and digestion of the PCR products was performed with the corresponding enzymes. The presence of mutations was confirmed by sequencing (Eurofins).

**Western blot analyses and quantification**. Overnight cultures of *P. aeruginosa* grown in LB were diluted to OD$_{600nm}$ of 0.05 and grown to OD$_{600nm}$ of 1 at 37 °C with shaking. Bacteria were harvested by centrifugation and pellets were resuspended in protein loading buffer and incubated 5 min at 98 °C. Samples were separated on denaturing 12 % polyacrylamide gels and transferred onto polyvinyl difluoride membranes. Purified MreC$_{Pa(36-330)}$ was used to raise polyclonal antibodies in rabbits following the manufacturer's recommendations (Biotem). Primary antibodies used for immunodetection were anti-MreC (Biotem; 1: 20,000 dilution) and anti-EF-Tu (Hycult Biotech #HM6010, 1:10,000 dilution), the latter having been used as a loading control. Secondary antibodies were anti-rabbit-HRP (Sigma #A9169; 1:20,000 dilution) and anti-mouse-HRP (Sigma #A9044; 1:20,000 dilution), respectively. Membranes were developed with the Luminata Classico Western HRP substrate (Millipore). Experiments were performed in three independent replicates. Quantification of the bands was performed using the ImageLab software (BioRad, version 6.0.1).

For comparison of the amount of the different MreC variants in the exponential and stationary phases, each MreC signal was first normalized to EF-Tu (used as loading control) and then compared to the intensity of wild-type MreC (taken as 100%). Statistical analyses were performed using SigmaPlot (version 11.0). For multiple comparisons, a one-way variance analysis (ANOVA) was performed, followed by Tukey's test. GraphPad Prism (version 7.04) was used for graph representation. Statistical significance was set at $p < 0.05$.

**Phylogenetic tree construction**. Amino acid sequences corresponding to 3,204 MreC variants in proteobacteria were extracted from the Uniprot website (https://www.uniprot.org). The alignment was performed using PROMALS3D and a Maximum Likelihood tree was generated using MEGA X[53], with 50 bootstraps. The final tree was visualized and annotated using iTOL v5[54]. Conservation of the positions of interest was extracted from the alignment in MEGA X and integrated into the final tree.

**Reporting summary**. Further information on research design is available in the Nature Research Reporting Summary linked to this article.

## Data availability
The atomic coordinates and the cryo-EM map were deposited in the PDB and EMDB, respectively, under the accession codes 6ZLV and EMD-11275. The final refined model coordinates and structure factors for MreC$_{Pa(97-258)}$ were deposited in the PDB under accession code 6ZM0. Source data are provided with this paper.

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

## Acknowledgements

The authors would like to thank Daphna Fenel for technical assistance, Emmanuelle Neumann for training on the microscope, Ambroise Desfosses for discussions on image analysis, Aymeric Peuch for help with the use of the EM computing cluster, Christine Ebel for advice on AUC data analysis and Peter Panchev for help with mutant construction and screening. This work was supported by a grant from the Agence Nationale de la Recherche to AD and IA (ANR-18-CE11-0019), including a postdoctoral fellowship to M.M.M. We also acknowledge support from the Laboratoire International Associé BACWALL (CNRS), and grants from the São Paulo Research Foundation (FAPESP) 2011/52067-6 and 2017/12436-9 to AD. Work in the Attrée team was partially supported by Fondation pour la Recherche Médicale (Team FRM 2017, DEQ20170336705). MJ-M received a Ph.D. fellowship from the French Ministry of Education and Research. FR-C was supported by FAPESP grants 2015/19906-5 and 2018/07148-7. CCM was supported by Ph.D. fellowship 88882.329485/2019-01 from CAPES. This work used the platforms of the Grenoble Instruct-ERIC center (ISBG; UMS 3518 CNRS-CEA-UGA-EMBL) within the Grenoble Partnership for Structural Biology (PSB), supported by FRISBI (ANR-10-INBS-0005-02) and GRAL, financed within the University Grenoble Alpes graduate school (Ecoles Universitaires de Recherche) CBH-EUR-GS (ANR-17-EURE-0003). The electron microscope facility is supported by the Auvergne-Rhône-Alpes Region, the Fondation Recherche Médicale (FRM), the Fonds FEDER, the Centre National de la Recherche Scientifique (CNRS), the CEA, the University of Grenoble Alpes, the EMBL, and the GIS-Infrastructures en Biologie Santé et Agronomie (IBISA). The IBS acknowledges integration into the Interdisciplinary Research Institute of Grenoble (IRIG, CEA). This work also made use of the MX2 beamline of the Brazilian Synchrotron Light

Laboratory (LNLS) and the ROBOLAB automatic crystallization facility of the LNBio, operated by the Center for Research in Energy and Materials (CNPEM) for the Ministry for Science, Technology and Innovation (MCTI, Brazil).

## Author contributions

A.M. and M.M.M. did large scale purifications of MreC$_{Pa}$ and A.M. performed negative-staining microscopy experiments. Cryo-EM data on MreC$_{Pa(36-330)}$ were collected by G.S. and the structure was solved by L.F.E. under the supervision of G.S. C.C.-M. and L.F.E. refined the pseudo-atomic structure derived from the cryo-EM structure. C.C.-M. solved and refined the X-ray crystal structure of MreC$_{Pa(97-258)}$. D.M.T. cloned MreC variants, crystallized MreC$_{Pa(97-258)}$, and collected X-ray data with C.C.-M. under the supervision of A.D. M.M.M., C.C.M., and F.R.-C. purified different MreC variants and performed AUC measurements; M.M.M. analyzed the AUC data. L.I. participated in protein expression efforts. I.A. supervised M.J.-M. and V.J. for construction of *P. aeruginosa* mutants, as well as the sequence and phylogenetic analyses performed by M.J.-M. Overall project supervision was performed by A.D. with input from I.A. and G.S. The manuscript was written by A.D. with input from all authors.

## Competing interests

The authors declare no competing interests.
