## [Peer Review File · Nature Communications]

REVIEWER COMMENTS

Reviewer #1 (Remarks to the Author):

In this article the authors shed new light on the structure and molecular mechanism of auto-association of MreC from the pathogen *Pseudomonas aeruginosa* and its role in regulating bacterial cell wall elongation. The authors probe the role of MreC using a variety of techniques including, analytical centrifugation, electron microscopy, X-ray crystallography, mutant analysis, and genetic methods. Phylogenetic analysis detailed in this manuscript suggests that the head-to-tail interaction of MreC monomers is widespread. Finally, the authors propose a model where the auto-association of MreC plays a role in regulating elongasome activity by sequestering MreC and reducing its binding to PBP2. The essential role of MreC for cell viability and its role in the putative switch mechanism regulating elongasome activity makes this work a significant, relevant and compelling contribution to our fundamental understanding of bacterial cell wall biogenesis.

Overall the experiments were well designed and the conclusions valid. However, the article could benefit from reworking in the areas mentioned below in order to make the article and its results/conclusions clearer.

General comment: Some discussion is warranted of potential caveats that may arise given the experiments are based on isolated filaments assembled *in vitro* as compared to within the context of their native environment and in the absence of their N-terminus.

Specific comment:

1. P.4, line 84: Negative stain EM of MreCPa shows tube-like oligomers but negative stain EM on MreC from *E. coli* and *A. baumannii* is stated to form "heterogeneous patches of different apparent sizes". These micrographs should be included with more effort to explain their form. Additionally, it would be helpful if the authors integrated these findings with their later claims in the paper that head-to-tail interaction of MreC monomers is widespread among rod-shaped bacteria.
2. P4, line 95 - Have the authors tested whether there is a concentration dependence governing the presence of the 3 species formed?
3. P4, line 98 - What specifically was the particle distribution? Given structures can be solved from minor populations, were there insufficient particles of the other forms to pursue their structures or were those species more heterogeneous/dynamic in conformation to preclude that analysis?
4. Pg.5, lines 116-118; With an overall resolution of 3.5 Angstroms and nice local resolution, it is somewhat confusing that the crystal structure was needed to trace the cryoEM map? Were there regions that were untraceable?
5. P.5, last paragraph: Regions 1, 2, and 3 defined here are shown to be broadly conserved across much of proteobacteria. However, only mutation of residues in regions 2 and 3 were found to influence the phenotype of *P. aeruginosa*. Explanation of possible reasons for the conservation of region 1 despite lack of apparent need for sequence conservation would be helpful.
6. Pg. 5 Some description of the nature of the interfaces and non covalent interactions between protofilaments would be useful.
7. P.7, line 161 - The region 3 mutation results in filaments with a diameter similar (from the figures) to the low concentration sample?
8. P. 7, the discussion spanning lines 176-197 is very speculative with limited experimental data to support. I would suggest this section should be as such toned down.
9. P.14: please include data acquisition mode (integrating, counting, etc.) for EM data.

10. P.25, Fig 1e and 1f: It would be helpful to have the inner/outer dimensions of the tube depicted within the figure.

11. Extended Data Table 2, line 142: Final Rfree of 31.84% is high for a 1.74Å structure suggesting that further refinement is needed and/or there are pathologies present in the data. Additionally, there is considerable spread between the Rfree (31.48%) and Rwork (22.10%). While it is understood higher resolution data was later collected and refined with more favourable R statistics, it is unclear why refinement statistics are included for the home source dataset when no model based on this data set was deposited.

12. Extended Data Table 2, line 126: In the Campinas data set the high-resolution shell is stated as 31.56-1.47 which appears to be a typo.

13. Extended Data Table 2, line 149: Ramachandran plot results incomplete for both datasets. Please include.

Reviewer #2 (Remarks to the Author):

The manuscript by Martins et al. presents cryo-EM and crystal structures of MreC-oligomeric complexes, providing evidence for head-to-tail and lateral interactions between the monomers through conserved interface residues, supported by the analysis of mutant proteins. Based on their data and the MreC-PBP2 co-structure previously published by the Dessen group, they propose a model of activation of PBP2 by MreC dimers, whereas MreC multimers are not capable of activating PBP2. In my view this work provides a major understanding in how MreC activates a key peptidoglycan synthase for cell elongation. I have only few minor suggestions about this interesting manuscript.

Minor points

1. Throughout the manuscript; the authors should consider to replace 'auto-association' by 'self-association' which, in my view, fits better to the MreC oligomers.
2. The authors should discuss the MreC oligomers with respect to the cellular copy number of the protein in *Pseudomonas* (or, if not know, the copy number for *E. coli*). It would be know for the readers how many MreC monomers are shown in the model in figure 1e, please add this number to the legend. Given the cellular copy number, how many of the exemplar tube shown in Fig. 1e would be present in an average cell?
3. Again on the tubular structure of the MreC oligomer (Fig. 1e/f): Is this architecture possible at all with MreC that contains the transmembrane helix? Can they model the MreC structure with TM helix and fit those monomers into the structure shown in Fig. 1e? Or would membrane-attached MreC molecules assemble oligomers of different structure?
4. Figure 1b. It is not clear what the insets show, please explain in the legend. Also, the axis in the inset curves should be labelled.
5. line 173. Please clarify what you mean by "cell wall length", length of a glycan chain or length of the sacculus?
6. line 195. Please provide the cellular copy number of MreC and indicate that this information applies to *E. coli*.
7. Model Figure 4f. Is there any experimental evidence to show that the MreC oligomers don't interact with PBP2?
8. Extended Figure 5. The 'loading control' (EF-Tu) actually shows significantly different amount of samples loaded on the lanes. For example for the mutant MreC there is very little EF-Tu signal in the 4 h sample. If EF-Tu is constant in samples, then the mutant MreC must be upregulated to high level after 4 hour chloramphenicol treatment. What is the explanation of this effect, and have they confirmed this result in replicate samples?
9. Extended Figure 5 panel e. Please add information to the legend to indicate whether or not there are statistically significant differences between the different MreC versions.
10. Extended Figure 6. Please explain the curves in the diagrams on the right side.

Reviewer #1 (Remarks to the Author):

In this article the authors shed new light on the structure and molecular mechanism of auto-association of MreC from the pathogen *Pseudomonas aeruginosa* and its role in regulating bacterial cell wall elongation. The authors probe the role of MreC using a variety of techniques including, analytical centrifugation, electron microscopy, X-ray crystallography, mutant analysis, and genetic methods. Phylogenetic analysis detailed in this manuscript suggests that the head-to-tail interaction of MreC monomers is widespread. Finally, the authors propose a model where the auto-association of MreC plays a role in regulating elongasome activity by sequestering MreC and reducing its binding to PBP2. The essential role of MreC for cell viability and its role in the putative switch mechanism regulating elongasome activity makes this work a significant, relevant and compelling contribution to our fundamental understanding of bacterial cell wall biogenesis.

Overall the experiments were well designed and the conclusions valid. However, the article could benefit from reworking in the areas mentioned below in order to make the article and its results/conclusions clearer.

General comment: Some discussion is warranted of potential caveats that may arise given the experiments are based on isolated filaments assembled *in vitro* as compared to within the context of their native environment and in the absence of their N-terminus.

We thank the reviewer for these positive comments and have extended our discussion to include a section that reads,

‘... Nevertheless, in our cryo-EM structure, all N-terminal regions point to a common direction in each protofilament (Fig. 2a), indicating a pattern in which MreC molecules could be anchored side-by-side on the bilayer. Notably, a comparable arrangement has also been observed in crystal structures of other MreC variants, from *Streptococcus pneumoniae*, *Listeria monocytogenes*, and *H. pylori*, where the N-terminal regions all point in a similar direction (18,19,20). Nevertheless, structures presented in this work were performed with purified proteins that assembled into oligomeric species *in vitro*, and thus it is conceivable that differences could arise in the context of the native environment of the cell.’

Specific comment:

1. P.4, line 84: Negative stain EM of MreCPa shows tube-like oligomers but negative stain EM on MreC from *E. coli* and *A. baumannii* is stated to form “heterogeneous patches of different apparent sizes”. These micrographs should be included with more effort to explain their form.

We have now included these micrographs as Ext Figs. 3e and f and have modified the text, both in the main section as well as in the figure legend, to better explain the forms of these structures.

Additionally, it would be helpful if the authors integrated these findings with their later claims in the paper that head-to-tail interaction of MreC monomers is widespread among rod-shaped bacteria.

In the crystal structures of MreC from *Listeria monocytogenes* (van den Ent et al., 2006), *Streptococcus pneumoniae* (Lovering et al. 2007), and *Helicobacter pylori* (Contreras-Martel et al., 2017), the respective N-terminal regions all point in a similar direction. This is also what is seen in this work. In the cell, this could favor anchoring of all individual monomers with the same directionality on the inner membrane, and could enable localized filament or patch formation. This idea is now further discussed on p. 8-9 (kindly refer to the response to your first general point, where we have reproduced the section of text that is now included in the manuscript).

2. P4, line 95 - Have the authors tested whether there is a concentration dependence governing the presence of the 3 species formed?

The three most stable species, observed in the cryo-EM images, were absent from low concentration samples. These tended to associate into short filaments resembling 'beads on a string', as seen in Ext. Fig. 3a. As samples were concentrated further, we started to observe a mixture of the three tube-like species described in the manuscript, but they did not present a distinct concentration-dependent profile, and could be identified in most of the samples that were visualized by negative staining.

3. P4, line 98 – What specifically was the particle distribution? Given structures can be solved from minor populations, were there insufficient particles of the other forms to pursue their structures or were those species more heterogeneous/dynamic in conformation to preclude that analysis?

Our structure was solved from the particle population that was in vast majority (the 200 Å diameter form, which represented 88% of the total particles). The two other forms only made up 3.3% of all particles. This is now clarified in the Materials & Methods section.

4. Pg.5, lines 116-118; With an overall resolution of 3.5 Angstroms and nice local resolution, it is somewhat confusing that the crystal structure was needed to trace the cryoEM map? Were there regions that were untraceable?

Indeed, there were some difficulties in the tracing of the N-terminal helix; in addition, the superposed structures of MreC variants from other bacteria (*L. monocytogenes*, *H. pylori*, *S. pneumoniae*) did not always agree in some of the loop regions. Hence, we opted to solve the high-resolution crystal structure of the *P. aeruginosa* MreC in order to avoid all doubts.

5. P.5, last paragraph: Regions 1, 2, and 3 defined here are shown to be broadly conserved across much of proteobacteria. However, only mutation of residues in regions 2 and 3 were found to influence the phenotype of *P. aeruginosa*. Explanation of possible reasons for the conservation of region 1 despite lack of apparent need for sequence conservation would be helpful.

Mutations introduced into Region 1 were somewhat conservative (proline to glycine, phenylalanine to alanine), since our objective was to decrease the hydrophobic character of the region without totally disrupting it. The absence of clear differences for MreC in both *in*

vitro and *in vivo* experiments could simply be a consequence of the fact that even the diminished hydrophobic character maintained in the region is sufficient to stabilize it.

6. Pg. 5 Some description of the nature of the interfaces and non covalent interactions between protofilaments would be useful.

The interface between the two protofilaments is dominated by charges between the β -sandwich regions of MreC. We have made an additional figure where the two facing dimers that form the representative tetramer were separated and rotated to face the reader; surface potential diagrams were also calculated. This analysis indicates that the dimers display charges that seem to be complementary, with those provided by Region 3 playing a prominent role. This figure has now been included as Extended Figure 5.

7. P.7, line 161 – The region 3 mutation results in filaments with a diameter similar (from the figures) to the low concentration sample?

The mutation in Region 3 (below, left) leads to filaments whose diameter is almost three times larger than that of the low concentration sample (below, right). This becomes more evident when one analyzes an enlarged image of a section of the structure (below, bottom right) where it is possible to notice that the Region 3 variant is composed of two fibers associated side-by-side (arrows). Most importantly, these forms are able to become highly elongated, which is not the case for the low concentration forms.

8. P. 7, the discussion spanning lines 176-197 is very speculative with limited experimental data to support. I would suggest this section should be as such toned down.

We agree with the reviewer and have now toned down the discussion.

9. P.14: please include data acquisition mode (integrating, counting, etc.) for EM data.

Data was acquired in counting mode, and this has now been clarified in the Methods section.

10. P.25, Fig 1e and 1f: It would be helpful to have the inner/outer dimensions of the tube depicted within the figure.

The dimensions of the tube depicted in Figures 1e and 1f have now been included in the figure legend.

11. Extended Data Table 2, line 142: Final Rfree of 31.84% is high for a 1.74Å structure suggesting that further refinement is needed and/or there are pathologies present in the data. Additionally, there is considerable spread between the Rfree (31.48%) and Rwork (22.10%).

Indeed, these values related to the data collected in-house, that had been employed only during the initial phases of the project. Values for the data collected later, at the synchrotron, show excellent statistics. As mentioned below, we have now removed values relating to the in-house data, and are only displaying values that correspond to the synchrotron data collection.

While it is understood higher resolution data was later collected and refined with more favourable R statistics, it is unclear why refinement statistics are included for the home source dataset when no model based on this data set was deposited.

We agree with the reviewer and have now included only refinement statistics for the dataset collected at the synchrotron.

12. Extended Data Table 2, line 126: In the Campinas data set the high-resolution shell is stated as 31.56-1.47 which appears to be a typo.

It was indeed a typo, that has now been corrected.

13. Extended Data Table 2, line 149: Ramachandran plot results incomplete for both datasets. Please include.

The Ramachandran results for the dataset collected at the synchrotron are included.

Reviewer #2 (Remarks to the Author):

The manuscript by Martins et al. presents cryo-EM and crystal structures of MreC-oligomeric complexes, providing evidence for head-to-tail and lateral interactions between the monomers through conserved interface residues, supported by the analysis of mutant proteins. Based on their data and the MreC-PBP2 co-structure previously published by the Dessen group, they propose a model of activation of PBP2 by MreC dimers, whereas MreC multimers are not capable of activating PBP2. In my view this work provides a major understanding in how MreC

activates a key peptidoglycan synthase for cell elongation. I have only few minor suggestions about this interesting manuscript.

Minor points

1. Throughout the manuscript; the authors should consider to replace 'auto-association' by 'self-association' which, in my view, fits better to the MreC oligomers.

We agree with the reviewer and have replaced 'auto-association' by 'self-association', including in the title.

2. The authors should discuss the MreC oligomers with respect to the cellular copy number of the protein in *Pseudomonas* (or, if not know, the copy number for *E. coli*). It would be know for the readers how many MreC monomers are shown in the model in figure 1e, please add this number to the legend.

In the red/green section of Figure 1e, 24 monomers are shown. However, the image only shows a section of the tube-like structure, which can extend to many microns; thus, the absolute number of monomers varies with the length of the tube.

Given the cellular copy number, how many of the exemplar tube shown in Fig. 1e would be present in an average cell?

In *E. coli*, the absolute rates of protein synthesis have been calculated in units of molecules per generation (Li et al. 2014). In the case of MreC, this estimation leads to ~740 molecules (in rich medium). However, we do not wish to suggest that the tubes themselves are present in an average cell, but that MreC employs the key interaction regions studied in this work to self-associate (to different levels), being thus able to exist in a PBP2-bound form and higher-order structures. In order to insist on the *in vitro* aspect of our tubular structures, we have now expanded the legend to Figure 1 to underline this idea. In addition, in the discussion, as well as in our model (Figure 4), we describe the different levels of MreC association as '... oligomers, short filaments, or patches', highlighting the importance of the interaction regions for their generation.

3. Again on the tubular structure of the MreC oligomer (Fig. 1e/f): Is this architecture possible at all with MreC that contains the transmembrane helix? Can they model the MreC structure with TM helix and fit those monomers into the structure shown in Fig. 1e? Or would membrane-attached MreC molecules assemble oligomers of different structure?

We believe that the MreC architecture that would be possible in the cell could resemble patches, or short filaments, that could resemble sections of the structure shown in Fig. 2a (which is a region of Fig. 1e). In Fig. 2a, we have indicated the potential position of the N-terminal regions, which would all be pointing in the same direction, towards the outside of the structure. Thus, we believe that the head-to-tail architecture would indeed be possible in the cell, being that the cryo-EM structure indicates that there would be enough space for TM regions to attach to the membrane without impediments.

4. Figure 1b. It is not clear what the insets show, please explain in the legend. Also, the axis in the inset curves should be labelled.

We have now labeled the axes in the inset diagrams and clarified their meaning in the legend to Fig. 1.

5. line 173. Please clarify what you mean by "cell wall length", length of a glycan chain or length of the sacculus?

Here, we refer to the length of the sacculus, but this could potentially be linked/regulated according to glycan chain length as well.

6. line 195. Please provide the cellular copy number of MreC and indicate that this information applies to *E. coli*.

The absolute value for the copy number of MreC in the cell depends on the medium in which the experiments were performed, according to Li et al (2014). For example, in the case of MreC, ~740 molecules were measured per generation in rich medium, while in minimal medium this value was ~ 180 molecules per generation. What is of interest is that the ratio of MreC to its partner proteins is maintained, independently of the medium tested. In the case of PBP2, ~320 molecules/generation were measured in rich medium vs ~80 in minimal medium, and for MreD, ~370 molecules per generation were measured in rich medium vs ~70 in minimal medium. Thus, we would like to refrain from mentioning absolute values in our text, but have included the fact that ribosome profiling experiments were done in *E. coli* (p. 8).

7. Model Figure 4f. Is there any experimental evidence to show that the MreC oligomers don't interact with PBP2?

[Redacted]

8. Extended Figure 5. The 'loading control' (EF-Tu) actually shows significantly different amount of samples loaded on the lanes. For example for the mutant MreC there is very little EF-Tu signal in the 4 h sample. If EF-Tu is constant in samples, then the mutant MreC must be upregulated to high level after 4 hour chloramphenicol treatment. What is the explanation of this effect, and have they confirmed this result in replicate samples?

The new extended Figure 6 (ex-Ext Fig 5) now shows the quantification of MreC levels normalized to EF-Tu from 3 independent immunoblotting experiments.

9. Extended Figure 5 panel e. Please add information to the legend to indicate whether or not there are statistically significant differences between the different MreC versions.

As mentioned above, we have replaced this figure with immunoblots and show the normalization of three MreC variants to EF-Tu.

10. Extended Figure 6. Please explain the curves in the diagrams on the right side.

We have now included a clarification within the legend to Ext Data Fig. 7 (ex-Fig. 6).

REVIEWERS' COMMENTS

Reviewer #1 (Remarks to the Author):

I believe the authors have answered the concerns from the initial review adequately.

Reviewer #2 (Remarks to the Author):

Most of my points about the previous version have been clarified in the manuscript. However, there are still two remaining issues that were not clarified:

1. Line 185: I believe the term "cell wall length" is misleading. If the authors refer to the length of the sacculus (or cell), then I recommend to change the wording in the text to "... once the appropriate cell length has been attained."
2. Extended figure 5 panel e. I appreciate that the authors added the diagram with the data from 3 experiments. However, the legend lacks information about what the error bars mean (SD or SEM) and there is still no statistical analysis of the data (p-value). Which MreC versions show a significant reduced level compared to wt protein? I also suggest to add the wt values (as black dots) into the diagram.

Responses to Reviewer 2

Reviewer #2 (Remarks to the Author):

Most of my points about the previous version have been clarified in the manuscript. However, there are still two remaining issues that were not clarified:

1. Line 185: I believe the term "cell wall length" is misleading. If the authors refer to the length of the sacculus (or cell), then I recommend to change the wording in the text to "... once the appropriate cell length has been attained."

This has been modified as per the reviewer's request.

2. Extended figure 5 panel e. I appreciate that the authors added the diagram with the data from 3 experiments. However, the legend lacks information about what the error bars mean (SD or SEM) and there is still no statistical analysis of the data (p-value). Which MreC versions show a significant reduced level compared to wt protein? I also suggest to add the wt values (as black dots) into the diagram.

We have now extended the figure legend to clarify issues on statistical analysis, p-values, and error bars (that indicate SD). As explained in the figure legend, mutant MreC values are reported in relation to those of wt MreC.